# The Associations Between Older Driver Licensure Laws with Travel and Passenger Behaviors Among Adults Aged 65 Years or Older (United States, 2003–2017)

**DOI:** 10.3390/ijerph18052251

**Published:** 2021-02-25

**Authors:** Sijun Shen, Marizen Ramirez, Cara J. Hamann, Nichole Morris, Corinne Peek-Asa, Motao Zhu

**Affiliations:** 1The Center for Injury Research and Policy, Abigail Wexner Research Institute at Nationwide Children’s Hospital, Columbus, OH 43205, USA; Sijun.Shen@nationwidechildrens.org; 2Department of Pediatrics, College of Medicine, The Ohio State University, Columbus, OH 43205, USA; 3Division of Environmental Health Sciences, School of Public Health, University of Minnesota Twin Cities, Minneapolis, MN 55455, USA; mramirez@umn.edu; 4Department of Epidemiology, University of Iowa, Iowa City, IA 52242, USA; cara-hamann@uiowa.edu; 5Injury Prevention Research Center, University of Iowa, Iowa City, IA 52242, USA; 6Department of Mechanical Engineering, HumanFIRST Laboratory, University of Minnesota, Minneapolis, MN 55455, USA; nlmorris@umn.edu; 7Department of Occupational and Environmental Health, University of Iowa, Iowa City, IA 52246, USA; corinne-peek-asa@uiowa.edu; 8Division of Epidemiology, College of Public Health, The Ohio State University, Columbus, OH 43210, USA

**Keywords:** gender difference, vision test, mandatory reporting laws for physicians, American Time Use Survey

## Abstract

Introduction: The aging population has been rapidly growing in the United States (U.S.). In line with this trend, older adults’ mobility and transportation safety are an increasing priority. Many states have implemented driver licensure laws specific to older adults to limit driving among the elderly with driving skill decline. Evaluations of these laws have primarily focused on their safety benefits related to older drivers’ fatal crash rate or injury rate. However, very few studies investigated licensure law effects on older adults’ mobility. Objective: The objective of our study is to evaluate the association between older driver licensure laws and older adult daily traveling and passenger exposure. Methods: The 2003–2017 American Time Use Survey (ATUS) data were linked with statewide driver licensure law provisions. Adults aged 55–64 years were used as the reference group to control for the effects of non-licensure-law factors (e.g., economic trend). We used modified Poisson regressions with robust variance to estimate the relationships between licensure law provisions and the likelihoods of older men and women’s daily traveling and passenger behaviors. Results: Laws requiring a vision test at in-person renewal were associated with increased daily traveling likelihood for women aged 75 years or older, primarily as a passenger. Laws requiring a knowledge test were related to a reduced daily overall traveling likelihood for women aged 75 years or older. Conclusions: In general, licensure law provisions are not strongly related to older adults’ mobility, in particular for older male adults. Older female adults’ daily mobility may be more likely to be influenced by the change of licensure laws than older male adults. The existence of gender-based disparities in responding to licensure laws requires future studies to account for the gender difference in estimating the effects of those traffic policies on older adults’ mobility and traffic safety.

## 1. Introduction

The United States (U.S.) population is aging. By 2030, one in five Americans will be 65 years or older [1]. The present aging population lives longer and is more active than their previous generations [2]. Due to these shifting demographics, older adult mobility is an increasing concern nationally. Reduced mobility for older adults could result in many negative outcomes, including a decline in independence [3], increase in mortality [4], depression [5], and adverse health conditions [6]. Driving remains the primary means for older adults to maintain their mobility and functional independence in the U.S. [7,8]. However, due to increased fragility along with physical and cognitive declines, older drivers have the highest rate of motor vehicle fatality and an increased risk of injury crashes per vehicle miles traveled [9,10].

To protect the health and safety of older adults and the roadway users who share the roadways with them, many states have implemented driver licensure laws specifically pertaining to older adults. It should be acknowledged that protecting older adults and other roadway users is the alleged purpose of the laws, but it is not necessarily reflective of research findings that older drivers pose the similar crash risk to others as middle-aged drivers [11]. The licensure law provisions include the licensure renewal period, means of license renewal (e.g., in-person, mail, or electronic), requirements for vision test, knowledge test, on-road driving skill assessment, and mandatory reporting laws for physicians. The licensure laws aim to identify older adults who may no longer be able to safely operate a vehicle and prevent them from continuing to drive on the road. Therefore, a competing public health issue rises between preventing unsafe older adults from driving on the road while promoting their mobility and independence. Stav [12] has argued that additional restrictions or tests for license renewal might result in fewer crashes and injuries for older adults, but they would also lead to older adults’ decreased mobility.

A number of previous studies investigated the safety benefits of older driver licensure laws by evaluating the impacts of the law provisions on older drivers’ fatal crash rates or injury crash rates [13,14,15,16,17]. Generally, vision tests, on-road tests, and more frequent licensure renewal cycles were not associated with reduced crash outcomes [13,14,15,16,17]. A shorter in-person licensure renewal was found to be related to a lower fatality rate among the oldest older drivers (aged 85 years or older) [15,16]. Other previous studies have estimated the association between licensure laws and older adult driving exposure [18,19,20,21]. The findings of those studies are mixed. Levy at al. [17] suggested that vision and on-road tests for adults aged 70 years or older may reduce their licensure rate, while McGwin, et al. [20] did not find evidence that vision test was a deterrent for adults 80 aged years or older to renew their license. Shen et al. [21] has identified that mandatory reporting laws for physicians were associated with reduced driving exposure for men 64–75 years and women 75 years or older. However, those studies only evaluated the direct impacts of licensure law provisions on older adults’ daily driving exposure and did not consider the indirect impacts of licensure laws on older adults’ non-driving activities. Some provisions of driver licensure laws may reduce older adults’ driving behaviors but result in their increased use of alternate choices for transport as a compensation for their reduced driving mobility. Shen et al. [8] has identified that riding in vehicles as a passenger is a major alternate transportation mode and, with increased age, older adults may regard riding in vehicles as a passenger as the main alternative to driving. Thus, it is very important to determine if licensure laws influence older adults’ overall mobility and their non-driving behaviors to better assist older adults preserving their mobility and promoting their physical and mental health.

No study, to our knowledge, has quantified the influences of licensure laws on the overall mobility for older adults and their daily riding behaviors. More importantly, very few studies have investigated if the licensure law provision may have different impacts on male and female daily mobility. Previous studies have identified that older women are more likely to stop driving than older men [22,23], and the effects of different licensure laws may vary between men and women on their decision to stop driving [21]. Thus, the objectives of our study were to characterize the relationships between various driver licensure law provisions and older adults’ daily overall traveling and passenger likelihoods and identify the potential gender disparities in responding to those licensure law provisions.

## 2. Methods

### 2.1. Dataset

Adults’ daily traveling and passenger behaviors were obtained from the 2003–2017 American Time Use Survey (ATUS). ATUS is a U.S. nationally representative survey collected by the U.S. Census Bureau that measures how U.S. residents 15 years or older spend their daily time on various activities (e.g., driving or watching TVs) [24]. The ATUS survey participants cover all U.S. household residents except active military personnel and residents living in institutions (e.g., nursing homes or prisons) [24]. Every survey respondent reported their daily activities between 4:00 am on the previous day and 4:00 am on the interview day. Each respondent’s data can be weighted to U.S. national estimates. With the assigned 160 replicate weights for each respondent, standard errors for those estimates can be computed via the successive difference replication (SDR) method [24].

For each activity, the ATUS respondents were asked to provide the starting and ending time of the activity, whether the activity was travel-related, with whom the respondent was accompanied during that activity, and where the activity took place. Thus, only activities coded as travel-related were included in our study. If a travel-related activity took place in privately-owned vehicles (POVs) where the respondent was a driver, that activity was coded as a driving activity. If a travel-related activity took place in POVs where the respondent was a passenger (including ride share services), that activity was coded as a passenger activity. POVs were defined as passenger vehicles, pickup trucks, and motorcycles/mopeds. Respondents who had at least one travel-related activity on their diary date were coded as traveling; those who had at least one driving activity were coded as driving; and those who had at least one passenger activity were coded as passenger.

The summarized daily traveling and passenger dataset for each individual respondent was then linked to state driver licensure laws effective on respondents’ diary date. State driver licensure laws were obtained from the law database used by Tefft [16] and online legislation databases [25,26]. Driver licensure laws were coded by two researchers with input from a third researcher for any discrepancy. The provisions of driver licensure law included in our study were the licensure renewal period, in-person licensure renewal period, vision test at in-person renewal, vision test without in-person renewal (a vision test report from health care providers if adults renew their license online or via mails), knowledge test, on-road test, and mandatory (vs. voluntary) physician reporting laws for medical conditions (which required physicians to report patients to the licensing authority under some medical conditions). Overall, every individual observation in our dataset characterized each respondent’s demographic information (age and gender), their diary date, and the licensure law provisions in effect on their diary date.

### 2.2. Statistical Analysis

The study population consisted of adults 55–64 years, 65–74 years, and 75 years or older. Adults 55–64 years were included as the reference group to control for the non-driver-licensure-law effects (e.g., gasoline price and unemployment rate) on travel-related behaviors over the study period. We selected adults aged 55–64 years as the reference group as they have the closest traveling patterns to adults 65 years or older and assumed that the unmeasured non-driver-licensure-law factors have similar effects across all the age groups, and the middle aged adults’ traveling behaviors were not highly influenced by driver licensure laws. We had tested if the employment status had various effects on the overall traveling and passenger behavior likelihoods among adults 55–-64, 65–74, and 75 years and found that, across all the age groups, having a job increased adults’ daily traveling and passenger likelihoods by the similar magnitude. In addition, we did not classify adults 85 years or older as an individual group, as in 2003 and 2004 ATUS top coded adults aged 80 years or older as “80”.

To estimate the daily traveling and passenger likelihoods for older male and female adults, two weighted Poisson regression models each with a natural log link function for each gender were developed. The modified Poisson regression model with robust errors (e.g., SDR) was an alternative to the log-binomial regression model due to the convergence issues associated with the later one [27,28]. The dependent variables were binary variables indicating if an adult traveled (0 = not traveled, 1 = traveled) or took a POV as a passenger (0 = not took a POV, 1 = took a POV) during their diary date. The independent variables were age category (coded as dummy variables for 65–74 years and 75 years or older, with 55–64 as a reference group) and provisions of the licensure laws, including licensure renewal period, in-person licensure renewal period, vision test at in-person renewal, vision test without in-person renewal, knowledge test, on-road test, and mandatory (vs. voluntary) physician reporting laws for medical conditions. The renewal period ranging from 1 to 12 years and the in-person renewal period ranging from 1–24 years were continuous variables in the models. It should be noted that some states (e.g., Arizona and Mississippi) did not require in-person renewal for adults in some age groups during our study period. For convenience of building models, we coded the length of the in-person renewal period for those states as 25 years. Additional sensitivity analysis was conducted to test the robustness of the estimates of in-person renewal period regarding how we coded the in-person renewal period for those unspecified states (25 years versus 30 years, 25 years versus 35 years, and complete removal of the observations with undetermined in-person renewal period), and results were shown in the Appendix A (Appendix A). The correlation between renewal period and in-person renewal period was 0.21, suggesting it was not likely to introduce multicollinearity issue into our models. The requirement of vision-test at in-person renewal, vision-test without in-person renewal, knowledge test, and mandatory reporting laws for physicians were coded as binary variables in the models (0 = absence of the provision, 1 = presence of the provision).

Our models also included the interactions between age-group indicating variables and licensure law provisions. The effects of licensure law provisions on a specific older adult age group were reflected by the estimates of their corresponding interactions. The non-licensure law effects on traveling and passenger behaviors were controlled by the main effects of licensure law provisions, and age effects were measured by the age group indicating variables in the models. As we used a natural log link function for each model, the estimates of regression coefficients represented additive changes of the log of likelihood, and the exponentiation of the estimates refers to multiplicative changes of likelihood. The exponentiation of the interactions was referred as relative risk ratios (rRRs) in our results which was interpreted as ratios of older adults traveling or riding in a vehicle with the presence of a licensure law provision versus the absence of the provision. To control the seasonality effects and yearly variation in adults’ travel-related behaviors, dummy variables for 4 quarters and 14 years (corresponding to each of the years of included survey data, 2003–2017) were also included in our models.

Additionally, during our study period, no state required adults in reference group (55–64 years) to pass an on-road test for license renewal, and thereby, the rRRs cannot be computed for this provision. Instead, we built separate Poisson models without the reference group to compute the risk ratios of on-road test for adults 75 years or older. The dependent variables for those separate models were binary variables indicating if an adult traveled and rode in a vehicle as a passenger. The independent variables only included the provision of licensure laws, year, and quarter. The Bonferroni correction was used to adjust for multiple tests on the two rRRs for each licensure law provision on overall traveling and passenger activities by gender and age group. Thus, the corrected significant level for our study is set as 0.025 (0.05/2). The *svy glm* function in Stata/IC 14.0 (StataCorp LLC, TX, USA) was used to build the weighted Poisson regression models, and the successive difference replication (SDR) method was used to compute the standard errors for the estimated regression coefficients.

## 3. Results

In total, 66,045 respondents were involved in our study, among which 27,202 were men and 38,843 were women. Adults aged 55–64 years, 65–74 years, and 75 years or older accounted for 44.3%, 31.2%, and 24.5% for the total respondents, respectively. About 6.7% respondents were classified as both drivers and passengers on the same day. Weighted to U.S national scale, the traveling proportions for male and female adults decreased as they aged (Table 1). However, men and women aged 65 years or older had larger proportions of passengers than their younger counterparts (Table 1). In general, within each age group, men had larger proportions of traveling but a smaller proportion of passenger than women (Table 1). As adults age, a larger proportion of women stop traveling than men (Table 1). Table 2 and Table 3 show the relative risk ratios (rRRs) for various licensure law provisions on older adults’ daily traveling and passenger behaviors. Since the corrected significant level in our study was set at 0.025 (0.05/2), the 97.5% confidence intervals (CIs) were presented in Table 2 and Table 3. For comprehensiveness, we also indicated the variables significant at an uncorrected significant level (0.05).

### 3.1. License Renewal Period

The median of renewal periods for adults aged 55–64 years, 65–74 years, and 75 years or older were 5.6, 5.4, and 5.2 years, respectively. For men 65–74 years, after controlling for temporal non-licensure-factor effects, a one-year reduction in licensure periods was not significantly associated with the likelihoods of daily traveling (rRR: 1.00, 97.5% CI: 0.99–1.01) and riding in a vehicle as a passenger (relative ratio: 0.93, 98.3% CI: 0.86–1.02; Table 2). The one-year reduction of license renewal period was not significantly associated with likelihoods of traveling and passenger for men and women 75 years or older or women aged 65–74 years.

### 3.2. In-Person Renewal Period

The national average in-person renewal period decreased with the increased age group. The median in-person renewal periods were 10.5, 9.4, and 5.9 years for adults aged 55-64 years, 65–74 years, and 75 years or older, respectively. Thus, older adults are required to renew their driver license more frequently than younger adults. For men and women aged 65–74 years and men aged 75 years or older, one-year reduction in the in-person renewal period was not associated with neither of the two travel-related behavior likelihoods. However, one-year reduction in the in-person renewal period was slightly but statistically significantly associated with a 0.3% reduction in daily traveling likelihood for women aged 65–74 years (rRR: 1.00, 97.5% CI: 0.99–1.00; Table 3). However, such a small reduction may not be associated with a practical difference. The sensitivity analysis presented in Appendix A shows that the coding of the undetermined in-person renewal period as 25 years has minimal effects on the estimates, as the point estimates in Appendix A were very similar to their counterparts shown in Table 2 and Table 3.

### 3.3. Vision Test

Over the study period, almost all the states in the U.S required drivers to take an on-site vision test when they renew their driver’s license at the license agency. Three states (Colorado, Florida, and Nevada) required drivers to submit a vision test report from their healthcare providers if they renew their license online or via mail. Overall, vision-test without in-person renewal was not associated with neither of the travel-related behavior likelihoods regardless of the gender or age group. The vision-test conducted at in-person renewal was significantly associated with increased daily traveling and passenger likelihoods for women 75 years or older (rRR: 1.10, 97.5% CI: 1.01–1.19; rRR:1.25, 97.5% CI:1.02–1.53; Table 3).

### 3.4. Knowledge Test

Five states required (California and Illinois) or had required (Indiana, Kansas, and Michigan), during a portion of the study, drivers above a certain age to pass a knowledge test while renewing their license in-person. California required drivers beginning at 70-years-old to take a knowledge test at in-person renewal, and Illinois required all drivers to pass a knowledge test when renewing their license, Indiana, Kansas, and Michigan repealed such law provision in 2004, 2010, and 2003, respectively. Knowledge test was not statistically significant for neither tested associations with the two travel-related behaviors’ likelihood for men and women aged 65–74 years. However, for women aged 75 years or older, knowledge test requirement was associated with almost 11% and 23% reduced likelihood in overall traveling and passenger behaviors.

### 3.5. Mandatory Reporting Laws for Physicians

In total, six states (California, Delaware, New Jersey, Nevada, Oregon, and Pennsylvania) and District of Columbia required their physicians to report to the licensing authority patients who were potentially not able to safely operate vehicles. The reporting criteria varied between states. Table 3 presents that mandatory reporting laws were associated with 21% increased passenger likelihood for women 75 years or older (rRR: 1.21, 97.5% CI: 1.00–1.45).

### 3.6. On-Road Test

Three states required (Illinois) or had required (Indiana and New Hampshire) drivers 70-year-old or older to pass an on-road test to renew their license. No states required drivers aged 55–64 years to take an on-road test when renewing their license. Thus, the risk ratios cannot be computed, and the estimated associations between on-road test and the two travel-related behaviors were confounded with other temporal effects. The on-road test was not significant among neither tested associations with the likelihood of overall traveling and passenger behaviors across the gender and age group.

## 4. Discussion

Our study was among the first studies to evaluate the relationships between driver licensure law provisions and older adults’ overall mobility (overall traveling) and passenger behaviors. We stratified our analysis by gender to account for the potential gender difference in responding to various law provisions. In general, adults’ overall traveling behaviors decreased, but their passenger behaviors increased as they aged. Within each age group, female adults had a smaller proportion of traveling but a larger proportion of being a passenger than their male counterparts.

We found that a one-year reduction in the in-person renewal period was statistically significantly associated with 0.3% reduction in daily traveling likelihood for female adults aged 65–74 years, but such a small reduction may not result in a practical difference. The reduction in the in-person renewal period was not associated with overall traveling and passenger likelihoods for men aged 65 years or older and women aged 75 years or older. Shen, Ratnapradipa [21] also found that a shorter in-person renewal period was not associated with older adults’ daily driving likelihood. Thus, in-person renewal period may have little effects on the older adults’ daily traveling, driving, and passenger behaviors. Previous studies have identified that more frequent in-person renewal reduced numbers of fatalities for drivers aged 85 years or older [15,16]. Together with our findings, shorter in-person renewal periods may limit the unsafe drivers aged 85 years or older from the road but preserve their mobility. In other words, our study suggests that the safety benefits of shorter in-person renewal periods for adults 85 years or older may not be at the cost of older adults’ mobility.

The requirement of a vision test at in-person renewal was associated with an increased traveling likelihood for women aged 75 years or older. The increased traveling likelihood was largely due to their increased likelihood of riding in a vehicle as a passenger. Women may differentially fail vision tests because they are more likely to develop cortical cataracts compared to men [29]. As a result, women may be more likely to choose to ride in a vehicle as a passenger if they are required to take a vision test at in-person renewal. However, the requirement of a vision-test without in-person renewal (adults are required to submit a vision test report from their healthcare providers when they renew their license online or by mail) was not associated with neither the overall traveling and passenger behavior likelihoods for older men and women. The insignificant results may be largely due to the limited sample size of observations subject to vision test without in-person renewal. Only three states (Colorado, Florida, and Nevada) had required vision tests without in-person renewal.

Related to the requirement of knowledge test, for women aged 75 years or older, the knowledge test was associated with their reduced daily mobility, and this reduced daily mobility was largely driven by their reduced passenger behaviors. It is possibly because older women who may fail to pass the knowledge test would also have concerns about their cognitive ability, resulting in voluntary reduction in overall mobility. However, Shen et al. [21] have suggested that knowledge test was not associated with older adults’ driving exposure. Policymakers should be aware that the knowledge test may have minimal impact on older adults’ driving behavior, but it may influence older female adults’ daily traveling decisions.

Mandatory reporting laws for physicians were not related to older adults’ overall mobility. However, the presence of mandatory reporting laws was associated with an increased passenger behavior likelihood for female adults aged 75 years or older. It was suggested that the mandatory reporting laws might reduce the driving likelihoods for women 75 years or older [21]. Thus, riding a vehicle as passengers may be regarded as an alternate transportation mode for women aged 75 years or older who were subjected to the mandatory reporting laws. However, as no state in our study period had ever enacted or repealed such laws, the comparisons of daily overall traveling and passenger behavior likelihoods in our study were cross-sectional and may be confounded with state-specific factors.

Our study identified gender disparities in responding to various licensure law provisions. Compared to older men, older women’s overall mobility and passenger behaviors are more likely to be impacted by stricter licensure laws, including the requirement of vision, knowledge tests at in-person renewal, and mandatory reporting laws. Although the underlying reasons for gender-based differences in the effects of licensure law provisions are unclear, policy makers should understand that older men and women might have different responses to the change of law provisions. Targeted interventions should be implemented to preserve the mobility of the most influenced older adult group, such as providing driving rehabilitation programs or improving the accessibility of alternative travel modes.

Our study has some limitations. First, we used the law in effect on each respondent’s diary date as the proxy of the effective laws on each respondent’s driver license. A respondent might renew their license before the implementation of the current renewal policy. Thus, they may be subject to the previous renewal policy rather than the one in effect on the diary date. The unavailability of the effective laws on respondents’ driver license may bias the estimates of the licensure law provisions. Second, our study did not account for the impacts of the enforcement manners of the licensure laws. The difference in enforcement manners with respect to strength, rigor, and difficulty may also influence the estimates of the law effects. Third, as no state required adults in the reference group to take an on-road test when they renew their license, the estimates of on-road test for men and women aged 75 years or older were confounded with other non-licensure-factors (e.g., gas price). Fourth, although our overall mobility includes any travel-related behaviors, we only modeled one alternate transportation mode (riding in a vehicle as a passenger) to driving for older adults in our mode-specific analysis. There are many other modes of traveling for older adults, such as walking, bus, and taxi. However, they accounted for less than 6% of daily trips for older adults. The limited sample size would result in highly skewed response variables (no trip on walking, bus, and taxi were dominated in the response variables) in our statistical models. Fifth, our study did not consider the interaction effects of licensure laws with geographical factors (e.g., urbanization). Adults living in urban areas have better access to public transportation and may respond differently to the same licensure law than adults living in rural areas. Future studies should continue to explore the effects of driver licensure laws on the mobility of adults living in urban and rural areas. Sixth, we used adults 55–64 years as the reference group in our study, assuming that the non-licensure-law factors have similar effects across all the age groups in our study. However, bias may still exist if some unmeasured non-licensure-law factors had various effects on adults’ traveling and passenger behavior likelihoods for adults in different age groups. Finally, as ATUS did not provide weather or other factors that may refrain their respondents from traveling, we could not further control the effects of those factors on individuals’ daily traveling or passenger likelihoods. However, as ATUS is a nationally representative survey, those individual factors may not severely influence the effects of licensure laws on national scale.

## 5. Conclusions

Overall, our study suggests that older men’s mobility was not strongly impacted by older driver licensure law provisions. Requirements for a vision test at in-person renewal was associated with increased overall mobility for older women. The increased overall mobility may be largely due to older women’s increase in riding a vehicle as a passenger. The knowledge test was associated with reduced overall traveling passenger likelihoods for women aged 75 years or older, respectively. With the presence of mandatory reporting laws, older women aged 75 years or older may choose to ride in a vehicle instead of driving a vehicle. Future studies should determine to what extent the safety benefits of the knowledge test and mandatory reporting laws are at the cost of older women’s mobility.

Our results provide additional evidence that gender disparities exist in older adults’ mobility and safety. When evaluating the effects of those traffic polices on older adults’ mobility or traffic safety, future studies should consider the potential gender difference in responding to those policies. Understanding the gender difference would also facilitate policymakers in optimizing their policy to protect older adults from traffic injuries and meanwhile preserve their mobility.

## Figures and Tables

**Table 1 ijerph-18-02251-t001:** Average daily traveling and passenger likelihoods by gender and age group, 2003–2017.

Adult Group	Travel ^a^			Passenger		
	Unweighted Count	Proportion (%)	95% CI ^b^	Unweighted Count	Proportion (%)	95% CI
Male						
55–64 years	10,807	84.6	83.7–85.5	1115	8.3	7.7–8.9
65–74 years	6612	78.3	77.2–79.4	817	9.3	8.5–10.1
75+ years	4005	70.4	68.8–72.0	663	11.7	10.6–12.9
Female						
55–64 years	13,229	82.3	81.6–83.0	4060	24.5	23.6–25.3
65–74 years	8898	74.8	73.9–75.6	3137	28.1	27.0–29.2
75+ years	6624	62.8	61.6–64.1	2626	26.6	25.5–27.7

Note: ^a^ Travel refers to all travel-related behaviors including driving, passenger, walking, bus, taxi, etc. ^b^ CI refers to confidence interval.

**Table 2 ijerph-18-02251-t002:** Relative risk ratios and 97.5% confidence intervals for overall traveling and riding in vehicles as a passenger for older men with men aged 55–64 years as the reference group, 2003–2017.

Law Provision	65–74 Years		75+ Years	
	Travel	Passenger	Travel	Passenger
Renewal period (one-year reduction)	1.00 (0.99–1.01) ^a^	0.93 (0.86–1.02)	0.99 (0.97–1.01)	1.07 (0.97–1.17)
In-person renewal period (one-year reduction)	1.00 (1.00–1.00)	1.02 (0.99–1.05)	1.00 (0.99–1.01)	0.99 (0.96–1.02)
Vision test at in-person renewal	0.96 (0.92–1.02)	1.05 (0.72–1.53)	0.96 (0.87–1.06)	1.35 (0.84–2.16)
Vision test without in-person renewal	0.96 (0.90–1.02)	1.01 (0.63–1.62)	1.03 (0.93–1.13)	1.07 (0.63–1.83)
Knowledge test	0.92 (0.84–1.01)	0.82 (0.46–1.49)	1.00 (0.89–1.11)	1.18 (0.65–2.15)
Mandatory reporting laws	0.95 (0.91–1.01) *	1.14 (0.81–1.61)	0.99 (0.90–1.08)	0.86 (0.59–1.23)
On-road test ^b^			1.04 (0.89–1.20)	0.83 (0.42–1.64)

Note: ^a^ Relative risk ratios (97.5% confidence intervals) were to compare daily traveling and riding likelihood for each provision of driver license renew laws adjusted for effects of non-driver-licensing-laws factors. ^b^ As there was no state that required adults in the reference group (aged 55–64 years) to take on-road test to renew their driver license, the risk ratio of the on-road test were computed by separate weighted survey Poisson model built on the data only including adults 75+ years. ***** indicates statistical significance at 0.05.

**Table 3 ijerph-18-02251-t003:** Relative risk ratios and 97.5% confidence intervals for overall traveling and riding in vehicles as a passenger for older women with women aged 55–64 years as the reference group, 2003–2017.

Law Provision	65–74 Years		75+ Years	
	Travel	Passenger	Travel	Passenger
Renewal period (one-year reduction)	1.00 (0.99–1.02) ^a^	0.99 (0.95–1.03)	1.02 (1.00–1.03)	1.00 (0.96–1.05)
In-person renewal period (one-year reduction)	1.00 (0.99–1.00)	0.99 (0.98-1.01)	1.00 (0.99–1.00) **	1.00 (0.99–1.01)
Vision test at in-person renewal	1.00 (0.95–1.06)	1.04 (0.85–1.26)	1.10 (1.01–1.19) **	1.25 (1.02–1.53) **
Vision test without in-person renewal	1.02 (0.95–1.10)	1.04 (0.83–1.30)	1.03 (0.94–1.12)	0.77 (0.59–1.02) *
Knowledge test	0.96 (0.89–1.03)	0.95 (0.77–1.17)	0.89 (0.81–0.97) **	0.77 (0.60–0.99) **
Mandatory reporting laws	0.98 (0.93–1.03)	0.91 (0.76–1.09)	1.02 (0.95–1.09)	1.21 (1.00–1.45) **
On-road test ^b^			1.02 (0.90–1.17)	1.04 (0.76–1.40)

Note: ^a^ Relative risk ratios (97.5% confidence intervals) were to compare daily traveling and riding likelihood for each provision of driver license renew laws adjusted for effects of non-driver-licensing-laws factors. ^b^ As there was no state that required adults in the reference group (aged 55–64 years) to take on-road test to renew their driver license, the risk ratio of the on-road test were computed by separate weighted survey Poisson model built on the data only including adults 75+ years. ***** indicates statistical significance at 0.05; ****** indicates statistical significance at 0.025 (0.05/2).

## Data Availability

Licensure renewal laws are available in the Appendix A], and the American Time Use Survey database are openly available in https://www.bls.gov/tus/.

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
