# Peer review of "The Associations Between Older Driver Licensure Laws with Travel and Passenger Behaviors Among Adults Aged 65 Years or Older (United States, 2003–2017)"

_ijerph, 2021, doi:10.3390/ijerph18052251_

Round 1

Reviewer 1 Report

Dear Authors,

Thank you for your work in this interesting area. I found your manuscript to be an interesting and well-written piece. I commend you for investigating the impact of licensure law effects on older people's mobility as it is an important area. 

My main concern with this manuscript is that it bears strong similarity to the publication by Shen et al. (2020) with limited acknowledgement of same, despite the fact that they have clearly come from the same data set, seem to have been analysed at or around the same time and with highly similar questions. The structure of the manuscript and even most paragraphs is almost identical and there was one sentence in the abstract that has been included almost verbatim. I have provided more extensive notes on this matter to the Editor for them to consider whether this degree of similarity is acceptable for this journal. 

Beyond those concerns, some other points for consideration are: 

  • Despite the similarities with the manuscript by Shen et al. (2020), this previous paper is cited only once. I think it would be beneficial for readers to have a greater sense of the distinction between this manuscript and the previous publication. What does this new manuscript add? If there is a specific reason why the content in the present manuscript was not included in the 2020 publication, can this be explained to the reader? Why is it important to know the difference between daily driving likelihood and driving likelihood, which seems to be one of the main differences between the two studies’ outcomes? More overt explanation of how this work builds on that of Shen et al. (2020) will help readers to avoid confusion.

  • 2, L56 58: The sentence starting, ‘To protect…’ could be interpreted as ageist and paternalistic, and may perpetuate misconceptions about older drivers. It might be worth acknowledging that although this is the alleged purpose of the laws, it’s not necessarily reflective of research findings, and that there are inherent biases in this reasoning. Recent research (Mitchell, 2018, doi 10.1108/S2044-994120170000010014) suggests that older drivers pose the same risk to others as middle-aged drivers.

  • The structure of the content around the aims could be refined so that no aims are introduced until all literature has been discussed (i.e., the material regarding Choi et al. 2013 and Feedman et al. 2002 should be discussed earlier in the introduction, before either of the aims or objectives are introduced).

  • In Tables 2 and 3, it’s unclear what the parenthetical material indicates – if these are confidence intervals as suggested in text, they should be presented as a range; e.g., “1.00 (0.99 - 1.01)” or “1.00 (0.99 to 1.01)” rather than separated by a comma. If I have misunderstood and these numbers represent something else, this should be provided in the table notes.

  • Results sections should be edited to avoid duplication of material found in tables.

  • Use of 55-64 as a reference group could be problematic because these individuals are largely working age, whereas those beyond this age are, in most instances, retired. I feel this is alluded to in the limitations section but could be explained more specifically.

  • The sentence at the end of Section 3.1 is grammatically incorrect.

  • There is a typographical error on p. 9, L374 and 377: “dairy date”

I wish you all the best for changes to your manuscript and thank you for your work.

Reviewer 2 Report

My only concern regarding this manuscript is that it presents data analyses based on the same dataset as in Shen, S., Ratnapradipa, K. L., Pervall, G. C., Sweeney, M., & Zhu, M. (2020). Driver License Renewal Laws and Older Adults' 518 Daily Driving, United States, 2003-2017. J Gerontol B Psychol Sci Soc Sci. doi: 10.1093/geronb/gbaa070 - and if I understood correctly, at least the analyses/results for "driving activities" (as opposed to "traveling" in general and "passenger activities") were actually already published within the aforementioned article (therefore, I put "yes" to the ethical concerns; apart from that, some theoretical and methodological content is also quite similar, but this is to be expected when writing on the same topic).

Although I understand the current manuscript as dealing with the ATUS data in more detail than the previous publication, I suggest clearly stating the fact and indicating what part of the analyses have been published already, and/or explaining in more detail the novelty/difference regarding the current analyses (besides, e.g., reporting median instead of average values). Better yet, I would recommend shortly summing up the published findings within the Introduction section, and only reporting the "new" findings within the results. The comparison thereof could then take place within the Discussion. This way, you could also avoid including some analyses that were already deemed impossible to compute based on the first article - e.g. on-road tests (due to non-requirements for 55-64 y.o.) - or are virtually the same as those published previously (e.g. mandatory reporting laws & driving likelihood for males and females - this part is practically the same as in Shen et al., 2020).

Round 2

Reviewer 1 Report

Dear Authors,

Thank you for your careful consideration of the feedback provided. I am satisfied with the changes made and congratulate you on your manuscript. 

Warm regards

Reviewer 2 Report

Thank you for all the revisions, the manuscript clearly benefited from the improvements.